# A Comparison of Pre-Service Science Teacher Education in Myanmar, the Philippines and Japan

**Wai Wai Kyi [1,*], Denis Dyvee Errabo [1,2]**  **and Tetsuo Isozaki [1]**

1   Graduate School of Humanities and Social Sciences, Hiroshima University, Higashihiroshima 739-8527, Japan; dderrabo@hiroshima-u.ac.jp (D.D.E.); isozaki@hiroshima-u.ac.jp (T.I.)
2   Department of Science Education, De La Salle University, Manila 0922, Philippines
*   Correspondence: d203664@hiroshima-u.ac.jp

**Abstract:** Teacher education is the very first step for preparing quality teachers and it is crucial to provide quality teacher training. This research aims to analyze and compare pre-service teacher education policies and programs in Myanmar, the Philippines, and Japan, focusing on secondary science teachers. A case study research design was employed by utilizing Technological Pedagogical and Content Knowledge (TPACK) framework. The result shows that TPACK components found in national education policies and programs in Myanmar were Content Knowledge (CK), Pedagogical Knowledge (PK), and Pedagogical Content Knowledge (PCK). On the other hand, TPACK found in the Philippines and Japan were CK, PK, Technological Knowledge (TK), PCK, Technological Content Knowledge (TCK), and Technological Pedagogical Knowledge (TPK). In all three countries, limited provision of Technological Pedagogical and Content Knowledge (TPACK) was found. It is interpreted that provision of a balanced and sufficient knowledge of TPACK is essential to well equip pre-service teachers with required knowledge and skills considering internationalization and transnational education. This research uncovered the general patterns and trends in pre-service teacher education for science in three Asian countries as well as their uniqueness and best practices.

**Keywords:** comparative study; pre-service; science teachers; teacher education; TPACK



## 1. Introduction

Teachers play an important role in determining the quality of the education system. They take the role in the interface of transmitting knowledge, skills, and values which are important for students to improve [1]. Accordingly, their quality is crucial and it has been globally accepted that it has a significant relationship with the quality of teacher education and students' learning outcomes [1]. There are a variety of benefits of quality teacher education: providing pre-service teachers with the opportunity to share their teaching practices [2]; updating teachers' knowledge with current teaching trends [3]; equipping teachers with the necessary teaching skills for their students' learning [4]; and improving teachers' qualifications [5]. While knowledge domains are considered to be important to teaching [6], teaching practice is also essential for pre-service teachers to learn to enact their educational theory/knowledge [7]. Teachers continuously develop and maintain their professional competencies starting with pre-service, through induction, in-service training, and on-going professional development programs [1]. Pre-service teacher education is the very first step for preparing quality teachers, and they could later continue their further professional development. This in turn highlights the need to design teacher education curriculum that best prepares teachers' skills and knowledge in this technological age in terms of national policies/standards for teacher education institutions, and programs/courses offered including teaching practice. Selecting qualified teachers is the beginning stage of teachers' professional development [8]. However, Lederman and Lederman (2015) [9] noted that more research is needed for unreported science teacher

preparation programs in many countries. Since preparing qualified teachers is crucial for their professional development, recent educational reforms have paid attention on enhancing high quality science teachers [10].

Considering the importance of pre-service teacher education to prepare qualified science teachers, the analysis of current teacher education systems is necessary to see if the programs are in accordance with the updated knowledge required of teachers or Technological Pedagogical and Content Knowledge (TPACK) in course work and/or practical work using Information and Communication Technology (ICT). The concept of TPACK recognizes that teaching with technology is more than just the technical aspects of using the technology, but also requires an understanding of how to use technology to support teaching and learning. Many studies have been conducted on multination comparative analysis of teaching education policies, standards, and/or programs. These studies employed TPACK framework in pre- and in-service teacher education in different contexts and different subjects using quantitative approach [11,12] and qualitative approach [13,14]. Additionally, researchers have shown an interest in examining the integration of technology into teaching for both pre-service [15,16] and in-service teachers [17]. To be specific, the integration of technology in science teaching-learning process is crucial because the utilization of technology can help concretize science concepts, and at the same time it can attract students' attention in those concepts [18]. It has also been argued that the development of TPACK in learning and teaching science can help teachers design and conduct experimental research to facilitate students' learning [19]. However, there is yet to analyze the science teacher education programs utilizing the TPACK framework, particularly the presence of TPACK from a tri-nation comparative approach with multiple case study design in three Asian countries. Through this research, unique practices as well as similar and different trends of TPACK in teacher education policies and programs in each system can be uncovered.

The current comparative analysis of teacher education systems in three countries is based on the following two conceptual groundings: internationalization and transnational education. On the one hand, the international or intercultural research and engagement especially in higher education (e.g., the provision of teacher education at the institutions/universities) could benefit not only the home country but also the foreign countries [20]. It also can have profound transformation by addressing the challenges by means of collaboration and engagement in research [21] such as comparative studies. This has become the major focus of policy and comparative studies [22]. On the other hand, globalization as a context of teacher education can also be relevant to exchanging or transporting of ideas of international, transnational perspectives on teaching, and teacher education [22]. Therefore, exploring the teacher education provision in each context, and comparing that with other countries in terms of common and different course offered could benefit the stakeholders including policy makers. This can help them see what is missing in their own contexts and what challenges could be addressed by collaborating with each other.

In addition, the comparative study of education helps us improve our understanding of our own past, locating ourselves more in the present time, and clearly discerning our educational future to some extent [23]. In this respect, we can reflect on our own past, and consider what could be improved to move forward. It is valuable as it helps us to fulfil our purposes of research in finding out, explaining, evaluating, critiquing, advocating, and developing our research interest [24]. Although we have to recognize that the findings from this case study in the field of social science cannot be utilized to generalize to other case studies, we show that comparative case study may provide an example of what occurs in other countries. Finally, according to Phillips and Ochs [25], researchers of comparative studies may be concerned with investigating educational issues and identifying procedures in other contexts to be adopted for the better provision in the home context. Cross-national studies have become important providing that globalization and internationalization have become the key themes of teacher education, leading to

evaluating the initial teacher education programs [26]. Therefore, research on teacher education from a cross-national approach has been conducted by researchers [27,28]. In this respect, it is worth investigating comparative studies on teacher education that have been done in the selected Asian contexts, and highlighting the gap to see what is yet to be done.

### 1.1. Justification of Selection of Three Countries

There are three major reasons for purposefully choosing Myanmar, the Philippines and Japan.

The first reason is the shared fact among three countries which is all three countries are from Asian region, with Japan being the developed country and the other two being the developing countries. Park [29] commented that international comparative researchers have regarded Japanese and Korean schools as having high quality teachers and students having equal access to highly qualified teachers [30,31]. The Organisation for Economic and Cooperative Development (OECD) argued that much can be learned from high performing countries in terms of offering a quality education for their students [32]. In this regard, it is worth exploring how science teachers in Japan are trained in their pre-service teacher education which ensures students have equal access to quality teacher education so that Myanmar and the Philippines pre-service teacher education can be compared with Japan's. The benefits will not only for Myanmar, and the Philippines, Japan can also learn what could be improved in their current system and what could be learned from their counterparts in terms of unique and best practices. Moreover, according to the Teaching and Learning International Survey (TALIS) 2018 report [33], Japan was above OECD average in receiving comprehensive initial education and training; assigned mentor for novice teachers; in-service teacher education for teachers and principals; feedback for teachers' understanding of their methods and practices; and collaborative professional learning for innovative and effective practices. Therefore, it is worth to learn how pre-service teacher education is provided in Japan.

The second reason is variation in administrative and political practices in terms of educational policies, decision making, and implementation. To be specific, in Myanmar, the whole education system is highly centralized and all higher education institutions including the teacher education institutions lack autonomy. As a result, the two universities of educations are required to strictly and exactly following the orders by Ministry of Education (MOE) with very limited flexibility in course provision in terms of variety based on the needs of specific contexts and depth of knowledge for high competency. However, in Japan and the Philippines, there are some levels of decentralizations. For example, Commission on Higher Education (CHED) in the Philippines and the Ministry of Education, Culture, Sports, Science and Technology (MEXT) in Japan, mandated the national policies for teacher education, and both private and national universities provide teacher education, allowing private universities having autonomy in terms of programs and provision as long as they follow the national level requirement. The advantage of this is that they have rooms for special, unique provision of courses and teaching practicums. This fact can be a crucial point in exploring how the administrative practices and power of decentralization and academic autonomy and freedom can allow for the unique and effective practices in teacher education.

Third, Japan is specifically selected to compare with Myanmar and the Philippines because of the support of MEXT and Japan International Cooperation Agency (JICA) to Myanmar and the Philippines for human resource development. To be specific, Myanmar and the Philippines are the recipients of Japanese government MEXT scholarships for teachers of under 35, and who graduated from university or teacher colleague [34]. In addition, Japanese Grant Aid for Human Resource Development Scholarship (JDS) has provided scholarships for Master's and Doctoral degrees for young government and non-government employees including teacher educators in teacher educators in teacher training institutions in many Asian countries including Myanmar and the Philippines [35]. Therefore, it is

worth exploring what kind of teacher training are being provided in teacher education institutions in Japan, and how MEXT mandated required credits for pre-service teachers.

### 1.2. Background of the Study

The recent technological age and global pandemic period has brought into focus the concept of TPACK which has interested many researchers and teachers. The selected three contexts are not exception and there have been a demand for integration of technology in education to meet the current needs and challenges that have resulted from COVID-19 restrictions on the face-to-face teaching in those contexts as explained in the following section. This section discusses the background of the education system and current issues in teacher education in the three selected contexts.

#### 1.2.1. Myanmar

In Myanmar, one of the priority areas of reforms to improve the quality of teaching [36] is enhancing the quality of pre-service teacher education by using a Teacher Competency Standard Framework (TCSF). However, this reform focused mainly on preparation for the shift from the 2-year Diploma in Teacher Education (DTEd) to the four-year Bachelor's degree for elementary and lower secondary school teachers, along with the reform of the teacher education curriculum with new content to link with the new basic education curriculum. In other words, despite the reform of the basic education curriculum and teacher education curriculum in Education Degree Colleges (EDCs), there is still a need for the reform of teacher education curriculum of Sagaing University of Education (SUOE) and Yangon University of Education (YUOE). This highlights the need to review and analyze the current teacher education curriculum of SUOE, one of the major providers of pre-service teacher education for BEd degree for upper secondary school teachers. In the Myanmar context, Keczer and Lay [37] conducted a comparative study on teaching competency standard frameworks for pre- and in-service teachers in Myanmar and Hungary. However, limited research has been conducted an in-depth analysis of TCSF and its integration in universities of education in Myanmar against the TPACK framework. In addition, there is a lack of comparative studies on current pre-service teacher education policies and programs for pre-service science teachers in Myanmar with those in other Asian countries.

COVID-19 has brought the closure of schools and universities in Myanmar. As a remedy for the closure of schools, Myanmar Digital Education Platform (MDEP) was established so that the public could have access to textbooks and supplementary materials. However, it was not utilized because of some hindrances such as no facilitation to use it, and no interactive engagement between teachers and students, and universities were unable to receive materials on the MDEP [38]. These facts highlight the need for integration of technological knowledge in basic education as well as teacher education sector. The quality teacher education system is necessary in Myanmar and the results of this study can be utilized for the improvement of Myanmar teacher education standard.

#### 1.2.2. The Philippines

In the Philippines, Higher Education Institutions (HEIs) play a very significant role as they are the ones responsible for the preparation of pre-service teachers who will be assigned in both the primary and secondary education sectors [39]. Improving the quality of education in the Philippines depends on the service of teachers who are adequately prepared for performing varied roles and functions. Therefore, designing the pre-service teacher education curriculum must consider higher standards in formulating the objectives, components, and processes [40]. Teacher Education Institutions (TEIs) are mandated to offer quality and holistic pre-service education to pre-service teachers. TEIs also provide theoretical and practical knowledge and skills on pedagogy. Analyzing the current teacher education policies and programs offered against TPACK framework and comparing those with other countries can give insight for improving teaching education system, which in turn improves the basic education.

In the Philippines, the Department of Education [41] mentioned that teachers must deliver quality and improved education, produce knowledge in their fields, integrating technology and results of other latest research into their teaching, and conduct research for continuous production and advancement of knowledge. Similarly, there is order from the CHED for all HEIs in the Philippines to deploy flexible and available leaning tools and delivery modes instead of campus learning [42]. The recent situation for students and lecturers regarding online learning has faced with many challenges [43]. This demands for teachers to be well equipped with Technological Knowledge (TK) and highlights the need to analyze the current pre-service teacher education policies and programs in terms of TPACK in teacher education. Previous research [44–51] has paid attention to education condition and issues in local contexts—i.e., the Philippines. Some studies on science teacher education, science teachers, and teaching in the Philippines are reported [44,52–54]. However, there is still a need to explore how pre-service science teachers are prepared in terms of content and pedagogy, as well as the training for integration of technology to deliver science education in the selected university.

### 1.2.3. Japan

In Japan, MEXT is responsible for pre-service primarily and supports in-service teacher education by law. The pre-service teacher education is provided by three major types of universities: National University, Municipal University, and Private University. Every university provides subjects required to acquire the licenses (certificates) for both lower and upper secondary school science teaching, and every prefectural board of education grants these licenses to those who completed the required course at the university [55]. Currently, there have been interrelated issues in school education and teacher education in Japan. For example, ICT usage in Japanese school was limited as Kihara [56] mentioned that before 2020, children had limited opportunities to use ICT in Japanese school classes according to the results of an international comparative survey. To address this, in Japan, "Global and Innovation Gateway for All (GIGA) School Concept" [56] (p. 4) budget was included as the supplementary budget and cabinet approved it in December 2019. The prefectural and municipal boards of education are responsible to improve the educational environment of schools, and yet there has been no progress nationwide until COVID-19 [56]. To remedy this situation, MEXT has pushed the above-mentioned GIGA school concept and now there are rapidly digitized educational environment in elementary, junior high, and high schools in Japan [56]. Therefore, it demands that initial teacher training provided in each university prepares a course for aspiring teachers to acquire the ability to use the ICT equipment in school lessons, and all aspiring teachers will acquire the related skills [56].

These facts show how Japanese government and schools are already considering the integration of technology in their education system, and how they already take the action to prepare the pre-service teachers to acquire the ability to use the ICT equipment. In other words, the idea of integrating TK in the current teacher education program has been started. From the perspectives of Myanmar and the Philippines, it is worth learning how and to what extent integration of technology is being provided in Japan. On the other hand, Japan can also make self-reflection on the provision of technological knowledge in pre-service teacher education program, and effective integration of it into their Content Knowledge (CK), Pedagogical Knowledge (PK), and Pedagogical Content Knowledge (PCK). TPACK is seen as an important component of effective teaching in the digital age [57]. In Japan, some comparative studies on teacher education between Japan and other countries have been conducted such as with the United States [58]; Hongkong [59]; Australia [60]; Australia and the United States [61]; and Turkey [62]. However, the comparative study with the selected two counties have not been conducted yet, especially for pre-service science teacher education. Studies on science teacher education in Japan are reported in the previous studies [55,63,64]. However, teacher education policy and programs in selected university in Japan, focusing on pre-service science teachers, has not been reported from a tri-nation comparative approach.

*1.3. Theoretical Framework: Technological Pedagogical and Content Knowledge: (TPACK) Framework*

The TPACK framework [57] was used to analyze the knowledge provided in three teacher education systems. That framework builds on Shulman's [6] concept of PCK. PCK is described by Shulman [65] as the particular content knowledge embodying the aspects of contents and the way subject is represented and formulated to make it comprehensible to others. TPACK encompasses three interrelated types of teacher knowledge important for professional practice: CK or the subject matter knowledge; PK or the knowledge of teaching and learning strategies; TK or knowledge about educational technologies and resources used to enhance student learning. A further four interrelated types of knowledge between these three core types are PCK, Technological Content Knowledge (TCK), Technological Pedagogical Knowledge (TPK), and TPACK. It is followed by the detailed elaboration of each component and criterion.

CK refers to the knowledge about a particular subject matter to teach (e.g., scientific concepts) [6]. To teach that particular subject, teachers need to deeply know the disciplines they teach [6,66] and develop CK [67]. Therefore, the criterion to analyze how pre-service teachers are provided with the subjects to develop CK is the presence or lack of subject matter knowledge specific to science subjects (e.g., physics and chemistry, etc.).

PK refers to teacher's knowledge about creating and facilitating teaching and learning environment where students can learn effectively [67]. That knowledge can be categorized into three: general PK, personal PK, and context-specific PK [68]. General PK encompasses teacher's knowledge about teaching strategies and models they will apply to teach a particular subject, and classroom organization and communication that can facilitate students' learning [66,69–72]. Personal PK is teachers' practical experiences and their personal beliefs and perceptions that affect their teaching [68]. Finally, context-specific PK is the knowledge which is formed by the combination of the other two: general PK and personal PK [73]. Therefore, this study explores the knowledge about those three types of PK that are provided for pre-service teachers to teach science subjects (e.g., physics and chemistry, etc.).

TK is concerned with the teacher's knowledge and skills necessary for using and mastering a variety of technological tools [66,69–71]. Through new and updated technology, TK is produced and adapted [67,70]. A teacher who possesses sufficient knowledge about technologies is able to choose and use appropriate technological tools to teach particular subjects, and he/she can constantly adapt to changing technologies [67]. In this study, the provision of TK in science teacher education courses (i.e., educational technology, utilization of technology in teaching science) in the selected contexts to equip pre-service teachers with TK will be explored.

PCK is teacher's knowledge about integrating or interacting between PK and CK [66]. PCK is distinctive in the sense that it can distinguish between an educator who educates students, and a content professional who is proficient in specific areas of content [65]. It means that, a teacher might be proficient in particular subject. However, he/she may not be able teach that content to students well. PCK means teacher's ability to apply appropriate instructional strategies (i.e., science teaching methods) to teach specific content (e.g., reflection of light) [67,72]. Therefore, this study investigates how selected teacher education policies and programs provide pre-service teachers with the opportunity to apply their PK to teach science subjects content (CK).

TCK is the knowledge that is developed by scaffolding within TPACK framework [67]. TCK is concerned with the teacher's knowledge and understanding of the effect of technology on particular subject area [66,69–72]. For instance, a science teacher with sufficient TK can determine the technologies that can be applied for explaining science concepts (e.g., measuring the speed of light, observing the nature of sound and the vibration of objects). Therefore, in this study, the provision of opportunity for pre-service teachers to integrate their TK to teach science subjects will be explored.

TPK is concerned with teacher's knowledge about selecting and determining the most compatible technologies for teaching and learning strategies to teach in particular

grades [70], and technologies that can contribute to particular education contexts in the best way [66,69]. For example, a science teacher with sufficient TPK knows which technologies would be most appropriate to effectively teach science concepts by considering the advantages and disadvantages of using different strategies such as graphs, videos, simulations, and power point slides. Therefore, this study explores how pre-service teachers are provided with opportunity to integrate their TK to teach science concepts by considering the nature of technologies that is most appropriate and effective.

Finally, TPACK is a framework that encompasses both understanding and defining what knowledge and skills will be required for teachers to effectively practice their PK in a learning environment where a technology is supported [72]. To be specific, it requires pre-service teachers to understand how their students can be facilitated by utilizing technology so that students' understanding of subject matter can be maximized [74]. It also includes practical experiences about the application of technology to teach science concepts [75,76]. For example, a science teacher with sufficient TK will utilize the most appropriate technology to teach the relationship between pressure, surface area and force (CK), and at the same time decide the most effective teaching strategies (PK) to teach science to best facilitate students learning about science concepts. This study investigates how teacher education policies and programs can train pre-service teachers to develop their TPACK with the creation of technology-supported environment with sufficient chances to select the most appropriate teaching strategies to teach a variety of science concepts.

### 1.4. Aim of the Study and Research Questions

This research aims to analyze and compare pre-service science teacher education in three nations, addressing two research questions as follows:

1. What type of knowledge is present in teacher education policies in the three countries under study?
2. How does the teacher education policy relate to the knowledge present in the teacher education programs in the three countries?

## 2. Materials and Methods

### 2.1. Research Design

A case study research design was employed along with the deductive content analysis technique to analyze and compare the content that is overtly presented in the teacher education policies and programs. The researchers aim to compare several cases and provide an in-depth understanding of the cases [77]. Replicable and valid inferences were provided by drawing from the existing data in order to provide knowledge, new insights, and representation of facts [78]. The policy documents were analyzed as they provide the intended rules and regulation of the pre-service teacher education. This comparative study on analyzing the education policy was conducted because research has the power to directly and indirectly influence the policy and practice in many different ways [79]. This research, therefore, analyzed the policy which reveals the power to situate ourselves and uncover the intentions and implementations. Conceptual framework was grounded on a tri-nation comparative study focusing on the policies and programs (Figure 1).

### 2.2. Data Sources and Sampling Strategy

The first set of data sources are the implementing TCSF documents mandated by the MOE in Myanmar, CHED in the Philippines, and MEXT in Japan. The second set is the implementing programs on the pre-service teacher education programs provided for university students who study physics, chemistry, and biology at Sagaing University of Education (SUOE) in Myanmar and at De La Salle University in the Philippines; and credits required for becoming upper secondary science teachers who study physics, chemistry, biology and earth science at Hiroshima University in Japan. In Myanmar, the program provided at SUOE was analyzed and it includes the courses and required number of credits for Bachelor's degree for pre-service teacher education at SUOE. In the Philippines, the

CHED Memorandum Order 75 series of 2017 for Bachelor of Secondary Education (Major in Science) was analyzed. As for the implementing Program, Bachelor in Science Education (BSED) Plan- BSED-PHY provided by De La Salle University was analyzed. In Japan, Educational Personnel Certification Act and Regulations- as the implementing Policies, overview of the number of credits required for teacher licensing [80] was utilized. The implementing programs for pre-service teacher education for upper secondary science in Hiroshima University was analyzed. To be specific, in Japan, teacher education institutions provide the courses following the required number of credits mandated by MEXT.

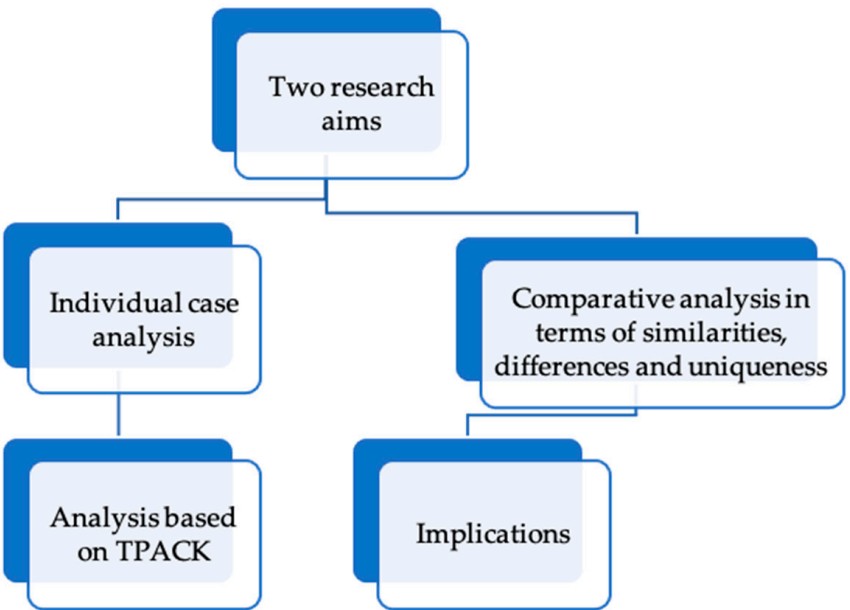

**Figure 1.** Conceptual framework for tri-nation comparative analysis. Source: Authors' own elaboration.

Firstly, the purposeful sampling was employed for the policies because the research aims to analyze the only national level documents in three countries. The sampling and data collection occurred from April 2022 to February 2023. Those documents were purposefully identified and selected because they can provide the rich information we need for this specific study [81]. The convenience sampling method or non-probability sampling method was employed for analyzing and comparing programs in the three selected universities due to the ease or convenience of accessibility of resources in the most proper way. This sampling strategy is helpful to identify and select information-rich cases considering the utilization of resources available to the authors, with each of them originating from these three contexts [81].

### 2.3. Units of Analysis: Sampling Units and Coding Units

This section explains the units of analysis: the sampling units and the coding units used in the study. The sampling units are the national level policies and programs provided at universities. The coding units are the written words, phrases, sentences in these policies and programs as suitable unit of analysis because letter, word, or sentence portion of pages can be the meaningful units (Robson, 1993), as cited in [82]. The documents were analyzed using TPACK framework and the predefined categories of analysis includes CK, PK, TK, PCK, TCK, TPK, and TPACK.

### 2.4. Coding and Content Analysis

Krippendorf's [78] coding using the content analysis was used as coding technique in developing replicable and truthful inference from the text to the context of their use. Deductive content analysis was utilized to answer the research questions: what type of knowledge is present in teacher education policies in the three countries under study?;

and how does the teacher education policy relate to the knowledge present in the teacher education programs in the three countries?

The five-step deductive content analysis was employed. First, deductive coding scheme was developed. Second, the authors coded the data especially deducting the knowledge provided in the policies and programs against TPACK framework. Third step is the reorganization and revision, in which researchers reorganized the coded data and revised after negotiating the discrepancies in the coding (i.e., components of TPACK found in documents) for an intra and inter-coding agreement. Fourth step is the preparation of the final codes. Final step is the presentation and comparison of results.

To ensure the validity of the data for deductively coding and analyzing documents, a coding scheme was developed and a sample coding scheme is provided (Appendix A). All generated codes are analyzed against the TPACK framework. For the reliability of data regarding reproducibility, the sampling units were analyzed against the TPACK categories by authors, with the same judgements being made for the same coding patterns [83]. A protocol was developed to ensure the reliability in terms of coding and interpretation of the manifest content: the surface information explicitly described in the documents. After coding, the authors discussed the interpretation jointly to confirm its accuracy. The results from each context were then compared.

## 3. Results

### 3.1. Type of Knowledge Present in Teacher Education Policies in Three Countries

This section discusses the results for the first research question obtained from the analysis of knowledge required of the national policies in three countries. The focus of analysis is CK, PK, TK, PCK, TCK, TPK, and TPACK. The data selected for this study are the policies and programs for science teacher education that trains pre-service science teachers. Therefore, the results presented below are related to science teaching. In addition, there are some "subjects" (which also mean "courses" in the Philippines case) specifically for science subjects while some are for both science major students and for other students.

#### 3.1.1. CK Present in Teacher Education Policies

Overall, it was found that CK is present in the teacher education policies in all three countries. To be specific, in Myanmar case, "Professional Knowledge and Understanding" category encompasses the knowledge required for teaching subject content (e.g., physics, chemistry) and it is concerned with CK. The Philippines's policies encompass general education (e.g., general psychology, English communication); "Elective Courses" (e.g., mathematics, science, and technology); and "Major Courses" (e.g., modern physics, electricity, and magnetism), and they are explicitly and directly related with the CK. Through learning General Psychology and English communication courses, pre-service science teachers relate what they have learned to apply this knowledge in teaching science content (CK). In Japanese mandated policies by MEXT, "Teaching Subjects" (e.g., physics, chemistry) in Japan are related with the CK. To be specific, the subjects offered under both liberal arts education and specialized education of program in science education cover CK.

#### 3.1.2. PK Present in Teacher Education Policies

In Myanmar, "Professional Skills and Practices" category is related to teachers' professional knowledge and understanding as well as teaching strategies (e.g., science teaching methods) for different educational contexts in accordance with the needs of individual students and different subject areas including science subjects. This shows that they are related to the PK. In the Philippines, "Professional Education Courses" (e.g., the Foundation/Theories and concepts, Methods and Strategies) are found to be related with PK because students who major in science learn these subjects and these methods and strategies are related with PK for teaching science. In Japan, "Subjects Related to Pedagogical or Teaching Contents and Teaching Methods" (MEXT, 2022) [80] are concerned with PK for teaching science.

### 3.1.3. TK Present in Teacher Education Policies

General education in the Philippines encompasses Science Technology and Society as one of Core Courses, and it was found to be related with the TK. The operation of information equipment in Japan is related with TK. In addition, MEXT mandates the integration of ICT in practical work in science subjects—physics, chemistry, biology, and earth science. In Myanmar, the explicit description for provision of TK is limited.

### 3.1.4. PCK Present in Teacher Education Policies

As discussed in TPACK framework section, pre-service teachers develop PCK means by applying appropriate instructional strategies to teach specific science content (e.g., reflection of light). In teaching practice, students have the opportunity to apply their PK to teach science concepts by applying their CK. In Myanmar, Professional Growth and Development requires all teachers including pre-service teachers to reflect on own teaching practice, and improve it. Therefore, they have to teach science by applying what they have learned (i.e., CK and PK), to develop PCK. In the Philippines, "Professional Education Courses": experiential learning including field study and teaching internship to teach science are related with PCK. In Japan, professional studies and subjects related to basic understanding of education, and a step-by-step teaching practice in Japan gives the chance to integrate CK and PK into PCK effectively. In Japan, especially in Hiroshima University, a systematic step-by-step teaching practice occurs in University's attached secondary schools from first year until third year [55]. During that period, it starts with introductory stage with the purpose of changing pre-service science teachers view from student view to teacher view [55]. The next year, student teachers are trained to engage in lesson study and their behavior are observed [55]. Finally, in third year, they engage in two-step practice- first is the three-day observation process about how their mentor works, and second is the engagement in teaching practice including lesson study in two different attached schools, lasting two weeks at each school [55]. It means that before teaching practice, pre-service teachers learn subject contents (CK) and pedagogical studies (PK), and they apply their CK and PK in their teaching practice. Therefore, it can be argued that they are provided with the opportunity to apply what they have learned (i.e., CK and PK) into PCK.

### 3.1.5. TCK Present in Teacher Education Policies

Regarding TCK presence in teacher education policies, unfortunately, Myanmar is limited in the provision of TCK. In the Philippines, pre-service science teachers are required to take educational technology course for teaching and learning science. Consequently, they are required to integrate their TK to develop TCK by practicing how to use technology in teaching science subjects (TCK). On the other hand, in Japan, MEXT strongly recommends that students who want to take teaching license to become a teacher takes "constitution of Japan, physical education, foreign language communication, operation of information equipment and data sciences" in liberal arts education by law. Therefore, "operation of information equipment and data sciences" requires the teacher education institution and faculties in university to provide pre-service teachers with the opportunity to learn ICT related subjects to teach subjects including science, and they must practice how to integrate ICT to teach a particular subject (i.e., science subjects) to develop TCK.

### 3.1.6. TPK Present in Teacher Education Policies

The results show that Myanmar lacks the provision of TPK. In the Philippines, pre-service science teachers learn educational technology courses, and they require to integrate their TK while employing particular teaching strategies PK to develop TPK to teach science subjects. In Japan, "operation of information equipment and data sciences" requires pre-service teachers to learn ICT related subjects to teach science subjects, and they must practice how to integrate ICT to teach particular science subjects by appropriately applying teaching strategies (PK) so that they develop TPK.

### 3.1.7. TPACK Present in Teacher Education Policies

TPACK can be developed through practical experiences regarding the application of technology (TK) to teach science concepts (CK) by utilizing their PK. In that respect, those aspects in all three contexts cannot be observed in the current analysis of overtly presented teacher education policies. However, the Philippines and Japan has the provision of TK, TPK and TCK, and TPACK could be observed in their teaching practice.

All these results can be summarized that, in Myanmar national teacher education policy, CK, PK, and PCK were found. On the other hand, the Philippines' and Japan's have provisions of CK, PK, TK, PCK, TPK, and TCK.

### 3.2. Type of Knowledge Related with in Teacher Education Programs in Three Countries

This section discusses the results for the second research question: teacher education programs for science education that train pre-service science teachers who specialize in the science subjects. The results presented as below are related with science teaching. The major results of teacher education policies in each country were presented and how these policies are related with in the teacher education program was examined. At the implementing programs level, the results were organized from the simplest level of knowledge: CK, PK and TK to the higher level of relating, applying, and integrating these knowledges into PCK, TCK, TPK, and TPACK.

Myanmar national teacher education policy mandated the inclusion of CK, PK, and PCK. The results from the analysis of the pre-service teacher education programs for science teachers in Myanmar show that throughout a 5-year program, pre-service teachers specialized in science must learn science subjects for CK according to their specializations (e.g., physics or chemistry), and teaching of science and science subjects and general methodology all of which focus on curriculum, pedagogy, lesson and unit plan, assessment, and preparation of teaching materials to teach science subjects for PK. They also have to learn compulsory education subjects such as Educational Psychology subject and Educational Theory and Management subject for PK and they learn and relate how to apply these skills to facilitate students learning in science. During and after learning both CK and PK, they also have to engage in step-by-step teaching practice from first year until fourth year which prepares them for PCK. During teaching practice in their third and fourth year, pre-service teachers have the opportunity to apply the appropriate teaching strategies they learned (PK), to teach particular science concepts (e.g., concave and convex lenses, pressure, and force). Unfortunately, students are not provided with the opportunity for TK, TCK, TPK, and TPACK. For the whole 5-year program, the subjects, however, are fixed and limited, and they do not have a variety of subjects (CK) to choose according to their requirements and interests. It can be generalized that there is provision of subjects for CK, PK, and PCK. However, there is a limited provision of TK, TCK, TPK, and TPACK.

The Philippines national teacher education policy mandated the inclusion of CK, PK, TK, PCK, TCK, and TPK. The results of analysis on knowledge provided by the university show that there are provisions of CK, PK, TK, and PCK. To be specific, "General Education Courses" (e.g., General Psychology and English communication) cover the knowledge of PK since pre-service science teachers have a chance to learn how to provide learning environments to facilitate students learning; "Major/cognate Courses" (e.g., Basic Electronics, Laboratory Physics) has enough provision of science concepts for CK; and finally, "Professional Education Courses" (e.g., Foundation of Educations, Curriculum, Teaching Strategies) including teaching practice to teach science cover the PK, PCK, and TK. However, General Psychology and English communication courses are provided for pre-service science teachers and they apply this knowledge in teaching science content (CK). In the Philippines, there is an immersion period at the partner school which provides students with the opportunities to observe the classes, document, and briefly reflect. It can be concluded that pre-service teachers are provided with the CK, PK, and the provision of opportunities to relate and integrate that knowledge into PCK. Similarly, pre-service science teachers have to learn educational technology courses and to integrate their TK in teaching

science subjects (CK) by utilizing their PK and develop TCK and TPK. To generalize, in the Philippines, CK, PK, TK, PCK, TCK, and TPK are provided, and TPACK is limited in the overt policy documents.

The Japanese national teacher education policy mandated the inclusion of CK, PK, TK, PCK, TCK and TPK. The subjects offered in Hiroshima University covers all these CK, PK, TK, PCK, TCK, and TPK under the provision of liberal arts education and specialized education including basic specialized subjects, specialized subjects, and specialized elective subjects. For example, students who belong to the department of science education learn that the specialized education for science education that includes basic knowledge and scientific literacy of, and practical and laboratory work in physics, chemistry, biology, and earth science, practical work in science education for CK. There are subjects on planning the teaching and learning activities, theories of science teaching-learning materials, evaluation in science education, theory on science curriculum for PK and PCK, seminar on science teaching and history of science teaching, etc. with a variety of choices for not only science subjects but also other elective subjects. The subjects that strongly requested by MEXT (e.g., constitution of Japan, physical education, foreign language communication, operation of information equipment, and data sciences) are provided in Hiroshima University in liberal arts education. Those subjects' credits can be counted as credits required for obtaining a teacher's license. Students who belong to the faculty of education and who want to take teaching certificates have to engage in step-by-step teaching practice from first year until fourth year which prepares them for PCK. There is practical work in physics, chemistry, biology, and earth science which include the application of ICT. Therefore, there is a provision of the opportunity to integrate ICT in CK and PK into TCK and TPK. However, there is a limited integration of the technological knowledge for teaching content knowledge with pedagogical knowledge which is TPACK. Therefore, in Japan, it can be generalized that the program covers CK, PK, TK, PCK, TCK, and TPK. However, the program is limited in its provision of TPACK which, however, can be observed in their actual teaching practice rather than in university lectures.

The overall results for analysis on TPACK knowledge in three countries' teacher education policies and programs can be interpreted that the type of knowledge in national policies are articulated in teacher education programs in three countries. The common trends among three contexts are CK, PK, and PCK, and limited provision of TPACK. Table 1 summarizes the comparison on overall results of the study. The criteria used for the analysis are based on the units of analysis that are elaborated and discussed in detail in theoretical framework (i.e., TPACK) section. As mentioned in the methodology section, the analysis was conducted based on the presence and lack of knowledge in the selected documents. A "√" is used for present of knowledge (i.e., explicitly and overtly present), "- "is used for lack of knowledge (cannot be found) and "Δ" is used for not explicitly present knowledge, but could be in teaching practice which can be clearly observed through class observation by the researchers.

**Table 1.** Comparison on TPACK in three countries' teacher education policies and programs.

| Units of Analysis | Myanmar | | The Philippines | | Japan | |
|---|---|---|---|---|---|---|
| | National Policies | University Programs | National Policies | University Programs | National Policies | University Programs |
| CK | √ | √ | √ | √ | √ | √ |
| PK | √ | √ | √ | √ | √ | √ |
| TK | - | - | √ | √ | √ | √ |
| PCK | √ | √ | √ | √ | √ | √ |
| TCK | - | - | √ | √ | √ | √ |

**Table 1.** *Cont.*

| Units of Analysis | Myanmar | | The Philippines | | Japan | |
|---|---|---|---|---|---|---|
| | National Policies | University Programs | National Policies | University Programs | National Policies | University Programs |
| TPK | - | - | √ | √ | √ | √ |
| TPACK | - | - | Δ | Δ | Δ | Δ |

Note: - Not found, √ Found, Δ In teaching practice.

## 4. Discussion

### 4.1. TPACK in Pre-Service Teacher Education

The overall results show that knowledge in teacher education national policies and programs in Myanmar has limited provision of TK, TCK, TPK, and TPACK. As summarized in Table 1, the Philippines and Japan are limited in provision of TPACK. However, it could be incorporated in teaching practice because in Japan, the course of department of teaching education aims to provide TPACK. The provision of TK, TCK, TPK in Japanese teacher education and integration of ICT in Japanese learning environment is because MEXT started the GIGA School Concept which enhances students and schools learning with ICT which can has a possibility to change student learning. This is because of the results of PISA 2018 which revealed the critical situation of ICT application in their learning environment [84]. To remedy this MEXT promotes the development of an ICT environment to bring out the potential of all children and realize high-quality learning by the Society 5.0 in Japan. Therefore, teachers should be required to develop ICT knowledge and skills both in pre-and in-service teacher education. This is one of the reasons that pre-service teacher education in Japan enhances development of TK, TCK, TPK, and TPACK rather than Myanmar.

This section discusses why TPACK framework should be considered and how it can be incorporated in the teacher education system to remedy the limited provision of TCK, TPK and TPACK in Myanmar, and TPACK in the Philippines and Japan. To be specific, it discusses how and what kind of teaching practice or practicum could be provided for pre-service teachers in three countries. Considering TPACK in pre-service teacher education is important and the use of technology for the improvement of teachers PCK is highlighted in the literature. As one specific example of TPACK study, Pamuk [14] (2012) examined pre-service teachers' development of TPACK and concluded that before teachers are able to integrate technology, they must prioritize their development of pedagogical content knowledge from their teaching experiences. It has been argued that the balance in studying many theories and practicing in real-world contexts is essential for becoming professional teachers [85]. Since this study uncovered the lack of TK, TCK, TPK, and TPACK, especially in Myanmar, the university could provide the educational technology courses, a variety of teaching method courses that are specific to particular contexts, and practicum. Pre-service science teachers can develop their TPACK by taking these courses and engaging with TPACK knowledge during these courses [86].

One of the best ways to relate theory and practice is the integration of CK, PK, and TK into PCK, TCK, and TPK in teaching practice leading to TPACK. This means that the programs should provide training with constructive alignment in the process of scaffolding from knowledge and comprehension of basic knowledge (i.e., CK, PK, and TK) through application, and relating to synthesis (i.e., PCK, TCK, TPK, and TPACK). When providing PCK through teaching practice, for example, it is important to provide the scaffolding and step-by-step practice beginning with teaching with support and assistance and progressing to independent practice that can finally lead to independent and effective teaching that utilizes TPACK. Much emphasis is put on learning pedagogical and content knowledge in close connection to practical learning for better preparation to teach [87]. Teaching practice is important as it is a dynamic combination of one's knowledge, skills, attitudes, values, and personal characteristics that empower the student teacher to act professionally and appropriately in a coherent way [88]. During initial training (preservice teacher education),

increasing PCK should be the primary aim of specialized courses in science education, which should be prominent in the initial stages of the training curriculum as well as learning from their own teaching experience [89]. Therefore, the authors suggested that three universities should provide the effective teaching practice that will give pre-service teachers the chance to apply and relate what they have learned: CK, PK, and TK into PCK, TPK, TCK, and TPACK.

In Japan, the main teaching practice period at the attached secondary school in the 6th semester is shorter than those in other countries such as the United Kingdom and Finland [55]. However, student teachers are effectively prepared even before they start their main teaching practice, and can be supported by university teachers and mentors [55]. The good relationship between the national university and attached or partnership school is one of key points to effective pre-service teacher education. In this way, they can have confidence when they teach that subject after graduation. This is not the case in Myanmar where pre-service teachers have to teach subjects that they have never encountered in their teaching practice. Myanmar can learn from the Philippines and Japan for the improvement of teacher education by providing pre-service teacher education with more subject related courses and extra elective subjects with a variety of choices for sufficient PK, CK as well as TK. They can also be provided with supportive and effective teaching practice for their specialized subject for improving PCK and additional subjects by collaborating with the schools. However, as the results highlight the limited provision of TPACK in all three contexts, the systematic and effective guidelines and mandates of integration of technology into teaching in the course as well as through teaching practice, for example, should be considered.

### 4.2. Uniqueness in Teacher Education in Three Countries

#### 4.2.1. Teacher Education in Myanmar

One uniqueness in Myanmar's teacher education system is that all first-year students have to learn "aspects of Myanmar" subject which introduces Myanmar language, culture and tradition. Despite their specialization (e.g., Science or Mathematics or English or Burmese Language), they need to learn that subject so that they are familiar with Myanmar language and they will be able to guide students.

#### 4.2.2. Teacher Education in the Philippines

Regarding the unique practices in each system, in the Philippines system, there are courses for provision of character formation and values development, Filipino language, English language, Philippines' culture, and national service training programs. We interpreted that it reflects consideration of the culture and tradition of the Philippines not only in school education but also in HIEs and teacher education.

#### 4.2.3. Teacher Education in Japan

Japan requires pre-service teachers for elementary and lower secondary school teachers to engage in the nursing, disabilities, and/or elderly people for seven days. In addition, after graduation, students can aim either to become upper secondary school science teachers or specialized personnel in businesses and public organizations or to proceed to Graduate School.

### 4.3. Similarities in Teacher Education in Three Countries

The trends show that there has been a provision of CK, PK, and PCK in all three systems. However, TK, TCK, and TPK are found only in the Philippines and Japan. The consideration and integration of TK with CK, PK, and PCK for TCK, TPK, and TPACK is limited in Myanmar. Another shared point between the Philippines and Japan is that the courses provided in universities cover the national requirements, with flexibility and variation in course provision resulting from the autonomy of the universities. This point

highlights how the political situation or the decentralization in sharing the power or autonomy in the university affect the knowledge provided for the students.

The next similar trend among three countries is the provision of CK, PK, and PCK and it shows that these types of knowledge are provided in all three systems. It can be assumed that these are the common knowledge. The courses offered at SUOE, De La Salle, and Hiroshima University equipped pre-service teachers well with the knowledge and practice they required in their teaching profession such as CK, PK, and PCK. As they are provided with both pedagogical knowledge and content knowledge which are central to effective pedagogies, pre-service teachers are be prepared well to represent content knowledge that they can grasp, anticipate difficulties, and build in support through explanations, relevant examples, metaphors, and actions [90]. It has also been revealed that in science teacher instruction, the key component of teacher knowledge is the CK and the PCK [91]. In addition, the instruction programs should provide learning opportunities in the way teachers can learn from experience in their training while having the opportunity to question their preconceptions about teaching and learning science [92]. In that sense, it is a good practice to equip pre-service teachers with enough CK and PCK. The increase in transnational education enables higher education institutions worldwide to promote curricula in their universities to the global market in which students from other countries can have access to diverse learning and teaching philosophies [93].

*4.4. Differences in Teacher Education in Three Countries*

The major difference among countries is that Myanmar's policies and programs are limited in TK while the Philippines and Japan has the provision of TK in both policies and programs. Regarding CK and PK, the Philippines and Japan provided a rich variety of choices especially in terms of CK. However, the sufficient provision of PK and PCK is yet to address. As Wang et al. (2016) [94] reported in the comparative study of preservice education for science teachers in five East Asian countries including Japan, one of two common challenges shared among countries is the balanced provision between CK and PK. The application of PK and CK is not only important but also a common issue in the pre-service teacher education. To remedy this, it is important to provide enough knowledge or the courses that can equip pre-service teachers with the competencies required for successful teachers. This fact is also in accordance with the results and suggestions of the study about enough and balanced PK, CK, TK and consideration of integrating them into PCK, TCK, TPK, and TPACK. Pre-service teacher education in Japan is well arranged in terms of the knowledge required by teachers as discussed by Authors [55], who commented that pre-service teacher education curricula are in line with all aspects of knowledge based on Shulman's [65] framework of seven types of knowledge. As discussed by Author [95], in pre-service training especially for science education, a variety of subjects together with subject studies and pedagogical studies should be offered as this training is crucial for providing opportunities for developing beliefs about aims and objectives of science teaching. Second difference among three contexts is the language of courses offered and medium of instruction. To be specific, in Myanmar, all teacher education courses (except for Myanmar language and aspects of Myanmar) are written in English, with the medium of instruction mainly in the language of Myanmar. In the Philippines, both the courses and medium of instruction are completely in English. Alternatively, in Japan, both the courses and medium of instruction are completely in Japanese. This significant finding shows one example of what Japan can learn from their counterparts as previously assumed in this research. Considering from the point of view of internationalization of education, Japan could provide some courses in English. In addition, teaching practice in the selected university in the Philippines is roughly four months for teaching practice including immersion, observation and actual practice period. Compared to that, Japan's main teaching practice is 4 weeks in the university's attached schools in the case of Hiroshima University at the third grade. Although Japan's is systematic and well-trained, Japan could also learn from the Philippines, where pre-service science teachers have more exposure to schools and teaching

which help them prepare more, immerse in schools, and practice. This finding is one of the advantages that the case study brings by revealing thing that is not expected in advance or in other case disconfirming researchers' expectations [96].

## 5. Conclusions

This study analyzed and compared teacher education policies and programs with a tri-nation comparative approach focusing on training for science teachers so that the three contexts are compared. TPACK components found in national education policies and programs in Myanmar were CK, PK, and PCK. On the other hand, TPACK found in the Philippines were CK, PK, TK, and PCK. In Japan, CK, PK, TK, PCK, TCK, and TPK were found. In all three countries, there was a limited provision of TPACK. Therefore, a balanced and sufficient provision of components of TPACK is suggested to provide to well equip pre-service teachers with the required knowledge and skills before entering their teaching profession. A similar trend between three countries is the provision of CK, PK, and PCK. The unique practices in each system are "aspects of Myanmar course" in Myanmar, courses for the provision of religious matters and national service training in the Philippines, and requirement for engagement with nursing, disabilities, elderly people for seven days for students who want to become elementary and lower secondary school teachers in Japan.

This research has implications in terms of internationalization, and transnational education. Internationalization can help pre-service teachers to become global citizens which could be referred to as a sense of belongingness to a global society or community and humanity for generating actions required to create a better future in a collective and civic way [97]. Exploring the presence and limited provision of knowledge required by teachers from cross national perspective may enable the policy makers and stakeholders design and implement better programs for future teacher education program. Especially, to remedy the limited provision of TPACK, it is suggested to provide the pre-service teachers with the integration of technology in content-specific situation so that they understand how to combine technology with science concepts (CK) and pedagogy (PK) [98]. In addition, it is suggested to use TPACK framework in preparing pre-service science teachers to integrate ICT in their teaching to increase their TK, TCK, and TPK as well as attitudes towards ICT as an effective tool for instruction [99]. This research also has the implications for pre-service teacher education for science in Asian countries to study the general trends and limited provision of components of TPACK in those regions.

This research is limited in the following ways. First, this research utilized only the mandated policies (intended) and university programs (implemented). It does not provide in-depth investigation of the outcomes of pre-service teacher education programs. Second, only the TPACK framework was utilized to analyze the policies and programs. Finally, this research focused on only three Asian countries, leaving a gap in investigation of other regions apart from Asia.

**Author Contributions:** W.W.K., D.D.E. and T.I. equally contributed to the study conception and design. Material preparation, data collection, and analysis were performed by the three authors. The first draft of the manuscript was written by W.W.K., D.D.E. and T.I. commented on previous versions of the manuscript and three authors revised until the final version. All authors have read and agreed to the published version of the manuscript.

**Funding:** This research received no external funding.

**Institutional Review Board Statement:** Not applicable.

**Informed Consent Statement:** Not applicable.

**Data Availability Statement:** Data presented are available upon reasonable request.

**Conflicts of Interest:** The authors declare no conflict of interest.

## Appendix A

**Table A1.** Sample coding scheme to map the policies and programs against TPACK framework.

| Units of Analysis | Myanmar | | The Philippines | | Japan | |
|---|---|---|---|---|---|---|
| | National Policies | University Programs | National Policies | University Programs | National Policies | University Programs |
| CK | Professional knowledge and understanding (teaching subject content such as physics, chemistry) | Science subjects (e.g., physics or chemistry) | Major Courses (e.g., modern physics, electricity and magnetisms) | Major/cognate Course (e.g., basic electronics, laboratory physics) | Subjects related to teaching contents (Subject studies) | Science subjects * in both liberal arts education and specialized education in program of science education |
| PK | Professional skills and practices (e.g., teaching strategies to teach science) | Pedagogy, classroom management, leadership, and educational philosophy related with science teaching | Foundation/. Theories and concepts, Methods and Strategies | Professional education courses (e.g., foundation of educations, curriculum, teaching strategies) | Subjects related to teaching methods (Pedagogical studies to teach science subjects) | Professional studies |
| TK | - | - | Science Technology and Society | Professional education courses (Education Technology) | Operation of information equipment | Information and data sciences in liberal arts education Provision of practical work in science subjects with application of ICT in specialized education |
| PCK | Professional skills and practices; Professional Growth and development to reflect on own teaching practice, and improve teaching practice | Teaching practice to teach science subjects | Professional education courses; experiential learning including field study and teaching internship to teach science subjects | Professional education courses (teaching practice to teach science subjects) | Professional studies; teaching practice (Pedagogical studies) Lesson study and teaching practice to teach science subjects | Step-by-step teaching practice and subjects related to teaching practice to teach science subjects |

* Science subjects cover physics, chemistry, biology, and earth science.

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
