# Peer review of "A Comparison of Pre-Service Science Teacher Education in Myanmar, the Philippines and Japan"

_education, doi:10.3390/educsci13070706_

Round 1

Reviewer 1 Report

The paper is well written and deals with an interesting comparison of policies and programs for pre-service education in Myanmar, the Philippines and Japan. A tool for this is a Technological Pedagogical and Content Knowledge framework called TPACK, an amalgamation of Shulmans concepts PK and CK and a third concept Technological knowledge of teaching (TK) which contains. knowledge of teaching about educational technologies and recourses used to enhance student learning.

Interesting outcomes of the study, with many relevant references. However, the Theoretical framework as Technological pedagogical and content knowledge could be described in a clearer way. The description of theoretical framework (1.3, pp 5, Figure 1) needs to be developed with focus on which kind of theoretical criteria are used in the analysis. This information is very important to understand Table 2 as well as the results. What makes the differences. I cannot find such information under Results 3.1.1 and 3.1.4.

The description of the theoretical approach TRACK needs to be developed with focus on the content of every part of Figure 1, related to the study as well as criteria used for the comparisons. The results, especially table 2 need to be more specific in order for explaining how TCK, TPK and TRACK are analyzed. This is not clear in the results and the description of policies and programs, and the theoretical framework gives no information of what conclusions and discussions are based on.

Author Response

We appreciate your thoughtful comments and valuable advices that are indeed helpful for the improvement of our manuscript. We deeply considered them, and carefully revised the manuscript.

The detailed revision in the main text was highlighted in “Blue color” for both Reviewer 1 and Reviewer 2. We used the same color highlight because almost all comments are similar in the sense that it is suggested to revise Theoretical Framework, Figure 1, Results, Table 2, and Discussion and Conclusion. However, we integrated some parts in Introduction section so that we introduce science teacher education and teaching from the beginning in the discussion of the manuscript, following Reviewer’s 2 suggestion. In addition, after revisiting the Data analysis, especially for TCK, TPK and TPACK, we have updated the results in the Philippines and Japan parts. Therefore, there are some updated parts in Abstract, Results, and Discussion too. They were highlighted in “Green Color”. We also submit our responses to clarify reviewers and table of corrections in the Responses/answers to reviewers’ comments file.

Reviewer 2 Report

The work presented assumes some relevance due to the theme addressed, within the scope of technologies and specific pre-service teacher education for science. The article reports an investigation of a multiple case study among three countries that have several synergies and particularities that make their selection understandable. The authors could have developed Shulman's work to explore each of the concepts addressed in this work and their contributions to teacher education. The literature review is adequate and justifies the arguments of the authors with relevance. However, I present some suggestions for improving the quality of the document:

(1) The study highlights pre-service teacher education in Japan as being of high quality, pointing to benefits of this research for Myanmar and Philippines improved their pre-service teacher education programs (101-103). However, in line 104, the authors emphasize that these benefits can also be for Japan (“Japan can also learn what could be improved in their current system and what could be learned from their counterparts in terms of unique and best practices”) . But it seems a somewhat contradictory statement due to the presentation of the high quality of training in Japan and the weaknesses of training in other countries, Myanmar and the Philippines. In lines 111 to 112 the authors state “Therefore, it is worth to learn how pre-service teacher education is provided in Japan”. With this study, does Japan actually have benefit for improving the quality of its training programs? These supposed benefits (for Japan) were not presented by the authors. Eventually, the speech could be, from the beginning, this premise, that is, what Myanmar and the Philippines can learn from Japan, thus adopting a position of unidirectional advantage: from Japan to Myanmar and the Philippines.

(2) Up to line 240, the reference to science teaching appears, but in a very tenuous way and since this is the focus of the article, it should be more present in the text of the article, from the beginning.

(3) In point 1.3 there could be a more in-depth and exploratory analysis of the scheme in Figure 1 that could contribute to an appreciation with greater understanding of the results.

(4) On line 321 the article mentions a figure (“Figure 3 shows”) that is not present in the text.

(5) In section 3 (line 341) some results need to understand their origin. For example, in 3.1.2 (PK present in teacher education policies), by the aforementioned category (“Professional Skills and Practices”), how did the authors conclude that “that they are related to the (…) PCK”? Do you refer that this category is present in different subject areas, but does it include the science area? It would be advisable to put some evidence that led the authors to take these conclusions.

(6) In section 3.1.4 I consider that these arguments are not related to the PCK, but to the PK. The PCK refers to the pedagogical knowledge of teaching the specific subject.

(7) What are the evidences for the assertion of line 393.

(8) Line 423: “General Education Courses” (e.g., general psychology, English communication, trigonometry) cover the knowledge of CK and PK - why does "general psychology" and "English communication" cover the the knowledge of CK in sciences ?

(9) Line 427: "Foundation of Education, Curriculum and Teaching Strategies" cover the PCK if they are specific to science teaching. It is the case?

(10) Line 472: “Japan is limited in provision of TPACK” but in table 2 it appears as found. Why do you write that it is limited? The same question for line 553.

(11) Section 5, Conclusions, lacks a link between the cited studies and the research conclusions. At this point there is reference to only one source that was cited for the first time at this point.

Author Response

(The authors gave the same response as above.)

Round 2

Reviewer 1 Report

This proposal is well-written and persuasively argued after revising. Very skillful use of relevant structure and the stringent content.

Reviewer 2 Report

The changes made to the document contributed to improving the quality of the article, as they clarified some identified doubts and provided a more accurate and comprehensive understanding of the content. The various parts of the article have benefited from the changes introduced. The final result is a more refined text, which conveys the study developed more effectively.